# Continuous Assessment of Mental Workload During Complex Human–Machine Interaction: Inferring Cognitive State from Signals External to the Operator [note 1]

**DOI:** 10.3390/s25123624

**Published:** 2025-06-09

**Authors:** Axel Roques, Dimitri Keriven Serpollet, Alice Nicolaï, Stéphane Buffat, Yannick James, Nicolas Vayatis, Ioannis Bargiotas, Pierre-Paul Vidal

**Affiliations:** 1Université Paris Cité, Université Paris Saclay, ENS Paris Saclay, CNRS, SSA, INSERM, Centre Borelli, F-75006 Paris, France; 2Training & Simulation, Thales AVS France SAS, 95520 Osny, France; 3Laboratoire GBCM, EA7528, CNAM, Hesam Université, 75003 Paris, France; 4Laboratoire d’Accidentologie de Biomécanique et du Comportement des Conducteurs, GIE Renault-PSA Groupes, 92000 Nanterre, France; 5CIAMS, Inria, Université Paris-Saclay, 91190 Gif-sur-Yvette, France; 6Plateforme d’Etude Sensorimotricité, CNRS UAR2009, Université Paris Cité, 75006 Paris, France; 7Machine Learning and I-Health International Cooperation Base of Zhejiang Province, Hangzhou Dianzi University, 310018 Hangzhou, China

**Keywords:** mental workload, machine learning, pilots, simulator

## Abstract

The use of complex human–machine interfaces (HMIs) has grown rapidly over the last few decades in both industrial and personal contexts. Now more than ever, the study of mental workload (MWL) in HMI operators appears essential: when mental demand exceeds task load, cognitive overload arises, increasing the risk of work-related fatigue or accidents. In this paper, we propose a data-driven approach for the continuous estimation of the MWL of professional helicopter pilots in realistic simulated flights. Physiological and operational parameters were used to train a novel machine-learning model of MWL. Our algorithm achieves good performance (ROC AUC score 0.836 ± 0.081, the maximum F1 score 0.842 ± 0.078 and PR AUC score 0.820 ± 0.097) and shows that the operational information outperforms the physiological signals in terms of predictive power for MWL. Our results pave the way towards intelligent systems able to monitor the MWL of HMI operators in real time and question the relevancy of physiology-derived metrics for this task.

## 1. Introduction

In recent years, the use of complex human–machine interfaces (HMI) has increased in several industrial fields, such as aeronautics, autonomous cars, nuclear power plants, etc. During such interactions, human performance can be measured by analyzing task load. Task load—also referred to as task demand or workload—is a term borrowed from the field of human factor that describes the amount of work that an operator should provide to complete a task [1,2]. Typically, task load encompasses three complementary dimensions: (1) the time required to complete the task, (2) the extent of information processing, and (3) the number of task switches when subcomponents of the task have a competing effect [3]. Mental workload (MWL) is defined as the subjective experience of a given task load. Indeed, for the same task load, factors such as time constraints, environment, or personal experience can induce varying MWL for different individuals [4]. From an ergonomic point of view, MWL is a dynamic concept that emerges from the interaction between the perceived task load and the perceived availability of the necessary resources to meet the task demand [5,6,7,8,9,10]. Accordingly, MWL can be seen as a continuum, whose instantaneous state depends on the ratio between demands and resources. At the lower end of the spectrum, the resources are perceived as more than sufficient by the individual to meet the needs of the situation: this is the under-load region. On the opposite end, they are perceived as insufficient: this is the overload region. In both cases, the MWL is suboptimal, which entails an increased risk of poor performance or even accidents [11].

Ensuring that the MWL remains within an optimal range is a critical issue when dealing with complex human–machine interfaces, otherwise, the risk of disaster could increase significantly. Case in point: the majority of accidents in aeronautics involve human factors (approximately 60–70% helicopter-wise) [12], indicating a loss of control over the HMI, the causes of which may be tied to under- or over-mobilization of cognitive resources. As for many complex human–machine interfaces, the technological advances in cockpits have gradually transformed the work of pilots into a multitasking job in which they manage several HMIs at the same time—piloting, communicating with air traffic control, monitoring multiple systems, displays and alerts, etc., [13]—such that effective attentional control, notably an efficient allocation of attentional resources, was found to be crucial for successful performance [14]. Hence, MWL implicitly lies at the forefront of the operational aspects of aeronautics, leading avionic experts to recommend monitoring the MWL in pilots [15].

Over the past few decades, several methods for assessing subjective mental workload have been developed, most of which are post hoc, self-reported, multidimensional analyses. Among the best-known assessments are the Cooper–Harper Rating scale [16], the Subjective Workload Assessment Technique (SWAT) [17], and the NASA-Task Load Index (NASA-TLX) [18]. Unidimensional scales such as Paas’ Subjective Cognitive Load (SCL) [19] can also be found.

This prompts the following question: how can MWL be quantified continuously? Despite the pervasive interest in its evaluation across a wide range of disciplines, a consensus on how to do so remains elusive. According to Rubio et al. [20], the operators themselves are in the best position to assess their own workload. Self-assessment measures have the advantage of integrating multiple intractable dimensions of MWL, such as effort, attitude, personality, satisfaction with the performance, etc. On the other hand, self-assessments of the MWL have several limitations. First and foremost, they are subjective indicators that may be biased due to unintentional under- or over-estimation of the MWL (e.g., based on idiosyncratic personality differences). Moreover, real-time self-assessments may distract the participant during a task. Despite these shortcomings, self-reports of MWL are sometimes the only feasible option when post hoc assessments are impractical or impossible (e.g., in the field, as they often require the individual to answer a questionnaire after task completion, such as the NASA-TLX) [20,21,22].

To address the limitations of self-assessments, extensive research has focused on the online monitoring of the MWL using various physiological signals (see [4,23,24] for a review). Rainieri et al. showed that during a simulated helicopter flight, pilots made shorter visual fixations on average during low-demanding tasks compared to high-demanding ones [25]. Rather coherently, heart-rate variability and electro-dermal activity variations were found to increase when pilots are submitted to a sudden high-demanding task such as engine failure [26]. More integrative approaches have been proposed to combine parameters derived from both physiological data and contextual data to improve the MWL estimation [27,28,29]. Such work demonstrated that MWL is better predicted by a combination of parameters than by a single one [27,30,31].

Faced with the diversity of signals that can be measured and the plethora of metrics that can be derived therefrom, data-driven approaches were deployed to investigate those that are most important for MWL quantification [27,28,32]. To give a recent highlight of such methods, Ding et al. found the breathing rate and the electro-dermal activity to be the most prevalent features to evaluate the MWL in computer-based tasks [33]. Among data-driven approaches, sequential training algorithms, e.g., random forest with mean decrease accuracy [34], sequential forward selection [34], AdaBoost classifiers [35]—or hierarchical weighted random forest [36] constitute two main classes of models that select a subset of relevant features during training to use for prediction. Several articles combined such data-driven algorithms with heuristics to infer prior knowledge into the machine-learning problem [37,38]. Others used physiological indices in addition to the NASA-TLX as ground truth to create a model estimating the mental workload of new car drivers; they found that the number of driving errors was positively correlated with the NASA-TLX score [39]. Finally, using a simple plane simulator, machine-learning classifiers, and EEG data, [40] concluded that machine-learning models have to be individualized to output sufficiently good predictions. Nonetheless, no continuous method of online, ecological measurement of MWL in the operators of complex HMIs has reached a consensus, notwithstanding the pressing need for appropriate tools to cope with the ever-increasing number and complexity of HMIs.

The goal of this paper is to offer new insights into this subject. Three key points were investigated: (1) the best possible performance of a simple machine-learning model to estimate MWL in ecological conditions, (2) the most relevant subset of signals to predict MWL in HMI operators, and (3) the best method to quantify MWL in the field.

To that end, nine professional helicopter pilots conducted two realistic scenarios in a Full-Flight simulator. Pilots were equipped with several sensors to monitor their physiological parameters. Operational data from the simulator were also recorded: i.e., the interaction between the pilots and the HMI and the state of the simulated aircraft. A heuristic-based bagging algorithm (HBagging [41]) was then used and trained on a collection of heterogeneous signals to estimate the MWL of professional helicopter pilots. The model can automatically select the optimal subset of metrics that leads to the best predictive power while discarding the others. Such an approach allowed us to address the challenge of ecological MWL prediction in an unbiased manner and to gain valuable knowledge about the most relevant sources of information for this task.

Surprisingly, our results show that operational information outperforms physiological signals in terms of predictive power for MWL. Overall, this study paves the way for intelligent systems capable of continuously monitoring the MWL of complex HMI operators in real time without disturbing them. They could, therefore, lead to adaptive online decision support systems to minimize the risk of incidents (e.g., automatic adjustments to displayed information) [27,42].

## 2. Materials and Methods

The Institutional Review Board Paris Descartes (CERES N°2017-35 dated 23/5/2017) approved the experimental protocols following the 1964 Helsinki Declaration. Data were collected from June 2017 to January 2018. Before testing, all participants gave their informed written consent. The authors did not have access to information that could identify individual participants during or after data collection.

An overview of the methodology is presented in Figure 1.

### 2.1. Population

Nine professional helicopter pilots (39.0±5.7 years old, all male) participated in this study. They had between 300 and 5000 (1934 ± 1295) hours of helicopter flights to their record and all had prior experience with the specific helicopter used in the simulator (Eurocopter EC135, 356 ± 219 h). Data collected in the first and second pilots were excluded due to faulty sensor measurements leading to high levels of noise.

Pilots were equipped with several sensors (Appendix A): a multi-purpose belt (heart rate and breathing rate, Equivital^®^, Cambridge, United Kingdom), an eye-tracker (SensoMotrics Intruments^®^, Teltow, Germany), and a head tracker (Thales Hobit^®^, Paris, France). All data were recorded and synchronized with an in-house software (Crew Monitoring System^®^). In addition to physiological data, all operational data from the simulator was recorded, i.e., helicopter state—trajectory, speed, altitude, orientation (roll, pitch, and yaw)—as well as information from the flight commands—displacements and force applied to the manipulanda, audio transmissions, etc.

### 2.2. Simulator

All experiments were conducted in a highly realistic, full flight, level D helicopter simulator (the highest level of certification, see Appendix A). It included a motion system (6 degrees of freedom motor-driven support surface) and high-fidelity visual surround that produced sensory cues similar to those experienced in an actual aircraft [43]. The experiments were preceded by a test flight to allow the pilots to familiarize themselves with the specificities of the simulator.

### 2.3. Flight Scenarios

Each pilot conducted two realistic scenarios (Figure 2). The first scenario was a typical recon mission. The pilot had to take off, fly off the coast to find a boat suspected of spilling oil in the sea, and then go back to the initial starting point. This reference scenario was the easiest and lasted about an hour. The second scenario involved more complex tasks. It started similarly but the pilots were promptly commanded, without previous notice, to perform a rescue mission, a “medevac”, i.e., a medical evacuation at sea. To do so, they had to change their flight plan to pick up a medical crew at a nearby hospital. With the medical crew on board, they then had to fly to the boat, land on the deck, and prepare for the return flight to the hospital. On the way back, the doctor demanded an emergency landing as the condition of the injured person was deteriorating. After this brief stop in very difficult weather conditions, the pilot had to take off again to finish their journey towards the hospital. The pilot then had to return to the original takeoff point, not without encountering an engine failure in the process.

The second scenario was specifically designed by experts to induce a high MWL; on the one hand because of the sheer context of the mission—which is inherently difficult and stressful—but also because additional elements were added to the scenario to trigger variations of MWL—bad weather conditions, instrument failures, etc. Each scenario was analyzed in detail through hierarchical task analysis by experts [44] to identify key moments in terms of MWL variations (e.g., ship landing, breakdown, emergency landing, etc.). Three MWL evaluation methods were compared.

Self-assessment: At key moments during both scenarios, the pilots were asked to self-report their MWL on a scale from 0 to 100, orally.

Experimental NASA-TLX: After completion of the second scenario, multiple NASA-Task Load Index (NASA-TLX [18]) questionnaires were used to assess at key moments of the mission six subcomponents of MWL: mental demand, physical demand, temporal demand, performance, frustration, and effort. Prior research has shown that the average of all sub-scores, called the Raw-TLX (RTLX), can be used to estimate the MWL [45].

Theoretical NASA-TLX: Two professional pilots and instructors also completed NASA-TLX questionnaires at the same key moments with their a priori predictions of the expected MWL. Their consensual estimates were collected to create the theoretically expected RTLX answers. 

### 2.4. Features

A total of $n_{total}=76$ features were computed. They were partitioned into three main groups: features derived from the physiology of the operator ($n_{operator}=23$), features from the state of the machine ($n_{machine}=28$) and features associated with the actions of the pilot on the flight instruments ($n_{interface}=25$). Details on the computation of each feature can be found in the Appendix A, and a summary of all features used in the model is presented in Table 1.

All physiological features were normalized. The goal was to ensure that potential idiosyncrasies in the physiological parameters would not prevent constructing a general model for the pilots. This method has proven to be valuable in accommodating inter-individual variations in MWL prediction [46]. In this work, physiological features were normalized by subtracting their mean value across the first scenario. In other words, each feature reflects the current variation of the metric relative to a personal, large-time-scale baseline.

To reflect human physiology [47], features were computed on 50 s windows starting 40 s before the auto-evaluation and ending 10 s afterward when the self-assessments were used as ground truth. With the experimental or theoretical NASA-TLX questionnaires, features were computed on windows of variable size that spanned the total duration of the task evaluated to reflect the temporal resolution inherent to the questionnaires.

### 2.5. Machine-Learning Model

The HBagging algorithm [41] was adapted to estimate MWL from multimodal sources of information. HBagging is a learning algorithm based on the combination of easily interpretable weak classifiers (unidimensional linear support vector machine), in which expert knowledge can be infused through heuristics. Thus, the HBagging algorithm inherits the full benefits of ensemble learning methods, namely robustness to noise and outliers, as well as interpretability [48].

Each weak classifier’s decision threshold is chosen such that the distance to the top left of the ROC curve is minimal (balance between true positives and true negatives, Appendix A). This requires binarizing ground truth MWL values into low and high states. To account for idiosyncratic variations in MWL subjective assessment, a personalized threshold was assigned to each pilot individually. This threshold was set in accordance to flight experts and computed as the 75th percentile of all MWL self-assessments reported during the first scenario. Responses to the theoretical and experimental NASA-TLX questionnaires were binarized according to the results of a recent meta-analysis on several dozens of papers using the NASA-TLX and RTLX [49]. Among the 152 entries for tasks involving “aircraft piloting”, the median answer was 47.78. This threshold was used to discriminate between low and high MWL evaluations. A visualization of this binarization process can be found in Appendix A.

Training was performed using 80% of the dataset using a custom cross-validation scheme such that, for each pilot, no MWL evaluation in the test set could be predicted using subsequent MWL reports. This scheme prevents the use of assessments that occur later in time to predict earlier values. During training, the model discards any weak classifier whose ROC AUC score falls below 0.6, to select only those features that have, by themselves, a slightly above chance to correctly classify MWL.

## 3. Results

### 3.1. Model Performance

The model was first trained with the complete dataset (i.e., the 76 features) and the subjective oral declaration as ground truth. The performance of the model was quantified on 1000 repetitions using the ROC AUC score (0.836±0.081), the maximum F1 score (0.842±0.078) and the PR AUC score (0.820±0.097). Table 2 details the model performance in terms of the ROC AUC score for each pilot. Our results show inter-pilot variations in the predictive power of the model, which suggests personal biases in the self-reports of MWL. We also observe relatively large AUC standard deviation values for each pilot (approx. 20%). This underlines the presence of idiosyncratic differences in the perception of MWL and its restitution, which prevents the model from generalizing well to all pilots at once.

Table 3 shows, in descending order, the features that were used the most frequently by the model. Our findings show a prevalence of metrics that capture the amount of interaction with the flight controls.

Following an initial training phase containing all available data points, a continuous estimate of the MWL experienced by each pilot was generated. Figure 3 represents two excerpts from such estimates that illustrate when the model’s output matches the ground truth and where it does not. An objective indication of task difficulty (Figure 3 grey highlights) is superimposed on the MWL estimations to better appreciate the robustness of the model. The figure suggests that the accuracy of the model decreases during tasks that cannot be accounted for by the input operational features.

### 3.2. Operational Features Outperform Physiological Parameters

To study the relative contribution of different sources of data, the model was trained with different subsets of features. The goal was to identify which category of signals possesses the most predictive power of MWL in ecological conditions. To ensure that meaningful feature sets were evaluated, the complete features set was partitioned into three main categories: (1) physiological data, (2) helicopter data, and (3) human–machine interface data. In each trial, the unselected feature sets were shuffled randomly, effectively turning them into nothing more than noise. The results can be found in Table 4.

We found that both simulator-related and HMI-related metrics outperform physiological data in terms of predictive power for MWL (by approx. 10% of AUC value). Furthermore, no singular subset achieves a performance equal to the first experiment, which suggests a lack of total redundancy between the three sub-datasets.

### 3.3. Impact of the Ground Truth on Model Performance

The impact of the ground truth used during training was investigated. The model was trained with the experimental RTLX answers as the ground truth using the complete features set. Sub-scores of the NASA-TLX—mental demand, temporal demand, frustration, and effort—were also considered, except for the physical demand, whose value never reached a ‘high’ state in all pilots, and for the performance, whose value rarely reached a ‘low’ state in all pilots. The same experiment was conducted with the theoretical RTLX. The sub-scores considered were the mental demand, the temporal demand, and the effort. The results can be found in Table 5.

Our findings show that using either of the averaged NASA-TLX scores leads to worse model performance compared to the self-assessments. A greater decrease can be observed in the scores of the theoretical NASA-TLX, which suggests that experts were not able to precisely anticipate the MWL experienced by the pilots. Regarding the sub-scores, only the “Mental demand” and “Effort” items led to results similar to those obtained with the self-reports.

## 4. Discussion

In this paper, we propose a machine-learning model for the estimation of the mental workload of helicopter pilots during a flight in a simulator. The model, based on previous works [41], offers good performance and provides operational experts with fully interpretable results. MWL was optimally predicted using a combination of operational, contextual, and physiological parameters, which demonstrates the importance of multimodal acquisition to best estimate the cognitive state of pilots in ecological conditions. Nevertheless, evidence suggests that the predictive power of the model remains high in the absence of physiological data.

The methods used in this paper are in line with state-of-the-art research in the prediction of mental workload using machine learning. Singh et al. [50] built a model to predict the MWL of naïve operators (but not pilots) from physiological recordings and achieved a score of 74.8% at best. Liu et al. [51] assessed the mental workload of pilots in training on a Boeing 737-800 simulator with the NASA-TLX questionnaire. Using a timeline analysis, they achieved a mean correct prediction rate of 56%. Both scores are substantially lower than those presented in this study. In contrast, our algorithm is simple, robust to noise and outliers thanks to a built-in feature selection process and an ensemble-learning approach, and fully interpretable (the decision threshold learned by the weak classifiers is straightforward).

### 4.1. Self-Reports vs. NASA-TLX Questionnaires for MWL Evaluation

Three different ground truths were used to train our model: the oral self-assessments of the pilots during the scenarios in the Full-Flight simulator, the NASA-TLX answered after the experiment, and the theoretical NASA-TLX completed by flight experts. Our findings show that the model obtained the best performance, in terms of the ROC AUC score, with the auto-evaluation. When the NASA-TLX was used, the scores were above chance but were not consistent across pilots (high standard deviation). These results are in line with prior works, which suggest that unidimensional estimation methods are better suited than multi-dimensional ones when performing a task, as they are less distracting and can be answered easily [52]. Nonetheless, some subsets of the NASA-TLX questionnaire were found to perform quite well when used as a ground truth. In particular, the sub-score “Effort” led to a performance similar to the self-reports. We believe that such results may indicate a poor understanding of the core concepts underlying each sub-score. “Effort” relates to an everyday concept that may seem intuitive to the pilots. On the contrary, sub-scores such as “Temporal demand” might appear confusing for some.

How come the experts’ evaluation—the theoretical NASA-TLX—led to inconclusive predictions of MWL? The experts may have had difficulties anticipating variations in perceptual-motor style across pilots, i.e., the stable idiosyncrasies that led to distinct ways of conducting the different tasks (see Vidal & Lacquaniti [53] for a review). Furthermore, the experts established the theoretical NASA-TLX scores without performing the scenarios themselves; they were not involved in the tasks and had no way of predicting possible fatigue due to the continuous operation of the helicopter, albeit simulated. The discrepancy between the expert’s opinion and the pilot’s experience is reminiscent of the mismatch between the assessment of pain by healthcare providers and their patients’ perception: physicians tend to underestimate the patient’s experience, especially when the pain increases [54].

### 4.2. Best Set of Signals for MWL Estimation

Of the nine most predictive features retained by our algorithm, only one was derived from gaze-tracking data (the proportion of time spent looking outside of the cockpit). Another feature ranked seventeenth: the gaze ellipse area (see Appendix A). This comes as a surprise, as previous research has shown the importance of gaze-tracking parameters in the study of MWL [55] such as the number of fixations, fixation duration, the number of saccades, saccade duration, saccade amplitude, and gaze distribution [5,56]. Although pilots use different visual strategies to respond to the specifics of the situations they encounter [57], the literature generally concurs that there exist unifying characteristics in pilots’ ocular behavior that reflect, for example, their level of expertise [58,59,60] and changes in mental workload [61]. The importance of the feature that tracks the time spent looking out of the cockpit in our study is consistent with these previous works. Singh and colleagues [50] reported that an increase in workload during a piloting task led to longer fixations on the external environment. In essence, the information retrieval strategies appear to be altered during high cognitive load situations: pilots excessively focus on the external environment relative to the flight instruments. This visual dependency is reminiscent of patients with poor postural and locomotor control who become overly dependent on vision and require rehabilitation to reduce this behavior [62].

In line with earlier works [50,63,64,65], two features derived from cardiorespiratory parameters were found to be predictive of MWL in our model (mean heart rate and mean inter-beat interval, cf. Appendix A). MWL influences the autonomic nervous system (ANS), which, in turn, regulates the heart rate [66]. Specifically, it was shown that heart rate increases and heart rate variability decreases with an increase in MWL [66]. The same trends were observed for the breathing rate [67,68], though the latter can also be affected by events unrelated to MWL, such as communication [68].

Nonetheless, overall, our findings suggest that physiological parameters are not the most effective for MWL estimation (Table 4). Multiple reasons could explain this lack of predictive power. First, as mentioned by Lorenzini et al., [69], most—if not all—of the studies on MWL that integrate physiological parameters are conducted in short-lasting laboratory-based experiments. Our study took place under more ecological conditions: professional pilots performed long flight protocols in a highly realistic helicopter simulator. This noisy and unpredictable environment may explain the lesser impact of gaze-tracking data and cardiorespiratory parameters compared to prior research as they can be affected by events unrelated to piloting. Second, previous studies have shown large inter-individual variations in perceptual-motor style during postural control [70] and locomotion [71] in controlled conditions. As such, we might expect sensorimotor style to have an even greater impact on the physiological features in realistic simulators.

In contrast, the most important features, in terms of predictive capabilities, were those derived from the interaction with the interface (the force and displacement of the manipulanda). To the best of our knowledge interface parameters had never been measured directly [69]. Often, the force applied by the operator on their devices is deduced from video recording of the body and not directly from the machine [72,73,74]. Our approach has two main advantages: (1) it is more practical from an operational point of view (no camera required); (2) it does not rely on complex three-dimensional modeling to reconstruct the human skeleton and structure, which is error-prone [72,73,74]. Among the most important feature used by the model was the standard deviation of the force applied to the pedals. Rotary-wing aircraft (helicopters) are notoriously more difficult to fly than fixed-wing aircraft (planes) because the pilot must constantly counteract the torque generated by the main rotor with the anti-torque pedals. It is, therefore, not surprising to see this feature at the top of the list. On a more global scale, the fact that interface-derived features are relevant for predicting MWL is consistent with mental workload increasing with the complexity of the maneuvers.

The significance of operational features in evaluating the internal state of pilots echoes another recent study [75]. In this paper, participants were seated at their workstations and asked regularly about how stressed they were while their heart rates, mouse movements, and keyboard entries were monitored. They found that stressed participants moved their mouse differently (more often and less precisely) and made more mistakes while typing on their keyboard. Interestingly, they concluded that mouse and keyboard (i.e., interface) data may better detect stress than cardiac data.

Since the movements of the manipulanda and the movements of the helicopter are coupled, the model also found that a number of features derived from helicopter motion were predictive of MWL. In a related paper (Roques et al., in preparation), we showed that pilots tend to avoid multi-dimensional movements and favor helicopter movements in a single plane of space. In this regard, the trajectory of the aircraft could be a powerful proxy for the mental load of the pilot, as multi-dimensional motions would tend to induce high MWL.

## 5. Limitations and Perspectives

This study presents several practical limitations. As with any study in naturalistic conditions, measurements of human behavior were noisy, impairing the quality of the extracted features. Due to equipment constraints, eye-tracking data were recorded at 60 Hz, a sampling rate that is sub-optimal to quantify saccadic eye movements (>200 Hz is preferred). Moreover, the number of subjects included in the experiment, while quite large when it comes to professional helicopter pilots, is rather small from a machine-learning perspective. This constraint is due to the great difficulty of obtaining confirmed pilots and the extremely high cost of conducting experiments under such realistic conditions. Despite the limited number of pilots, the scenarios were long, with numerous self-reports of MWL during the flights. These auto-evaluations are nevertheless a perturbation that, if too frequent, may be detrimental to the piloting task. Recent work focused on data labeling may provide solutions to this problem. Zhang et al. [76] used machine-learning techniques to leverage these large portions of unlabeled data to improve instantaneous mental workload prediction. Such approaches appear to be perfect candidates for the present context.

The range of potential sensors that could be used was limited by constraints related to the pilot equipment. As such, several sensors that are commonly found in the mental workload literature—e.g., electroencephalogram, galvanic skin response, and electromyogram [77]—were not employed. Additionally, the complete set of features computed from the sensors used was, of course, not exhaustive. For instance, promising frequency-based features based on the electrocardiogram [77] were not used in favor of more typical features.

The use of a simulated environment, albeit highly realistic, may also question the reliability of the results. In aviation, experiments in real flights are prohibited due to safety concerns and most research is typically conducted in a simulator. The particular simulator that was used for this study was one of the best in the world in terms of realism (with respect to sensory inputs and flight characteristics) and thus pilots were really immersed during the scenario. It is, therefore, reasonable to expect similar behavior between the simulated flights and real flights.

Finally, the results presented in this study offer limited comparison to other machine-learning methods. A larger study was carried out in which different types of algorithms were tested. However, this work is beyond the scope of this article, which is focused on analyzing the result of a simple yet effective model rather than achieving the best possible MWL estimation with a complex, custom algorithm, but more specifically analyzing the results of a simple yet effective learning algorithm. To this end, the HBagging algorithm was used. It provides robust and interpretable results: using an ensemble of simple classifiers (linear SVMs), the best features for MWL estimation could easily be accessed from the model along with the decision threshold used to classify the MWL as “low” or “high”. Such knowledge is essential in the particular context of aviation where risk is high and the causal links between flight events and the cognitive state of the pilot are still poorly understood. Nevertheless, the HBagging model presents several limitations. The simple decision threshold learned by each weak classifier may miss complex, non-linear correlations between the feature and the MWL. For the same reason, the heuristics used in the model are also limited. Furthermore, all features are considered independently. This approach was adopted to address the limited data available and their heterogeneous nature, which precluded the use of more complex models without compromising robustness and performance. With a larger dataset, a tree structure would potentially better capture the complexity of mental workload more effectively and take advantage of the multimodal nature of the feature set.

The findings outlined in this paper could be investigated using control theory. If we look at the chain of command for controlling a helicopter, the pilot is upstream (they make decisions and take actions) while the machine is downstream (it moves according to the pilot’s impetus). The fact that our predictive model of mental workload relies almost exclusively on downstream features suggests that the chain of command acts as a filter. Upstream features, such as those derived from physiological readings, do not provide an accurate account of mental workload in an ecological setting because they are the product of a complex assortment of intractable factors coupled with strong interpersonal variability. In contrast, downstream features appear more reliable in predicting mental workload, as if the aforementioned “noise” was eliminated as one moves down the chain of command. This hypothesis could make the recording of physiological data—which is problematic from a practical, ethical, and theoretical point of view—less crucial when it comes to MWL estimation.

Overall, this study paves the way for a new way to assess the MWL of complex HMI operators. The generalization of such a process in everyday life (e.g., for drivers of manned vehicles, operators of nuclear power plants, etc.) could lead to an improvement in collective safety in a society facing a proliferation of increasingly complex HMIs, fueled in part by the rise of artificial intelligence.

## Figures and Tables

**Figure 1 sensors-25-03624-f001:**
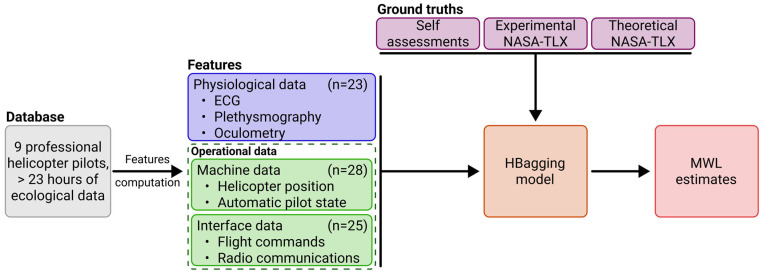
Overview of the proposed methodology.

**Figure 2 sensors-25-03624-f002:**
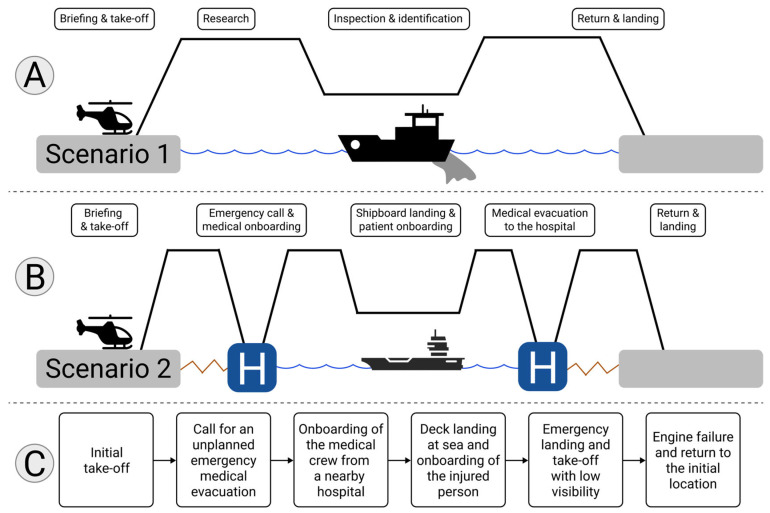
Flight scenarios. (**A**,**B**) Illustration of the different phases of the two scenarios. (**C**) breakdown of the second scenario’s unexpected events.

**Figure 3 sensors-25-03624-f003:**
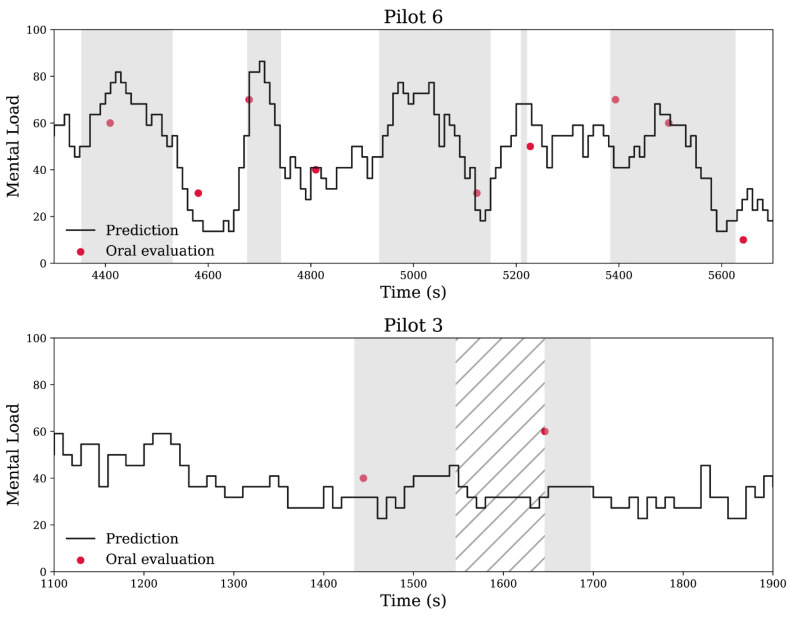
MWL estimation in two examples. Excerpts of the continuous mental workload prediction for the sixth pilot (above) and the third one (below) in the second scenario. The red dots correspond to the self-assessments of the mental workload (the ground truth). Portions of the scenarios highlighted in gray represent flight phases where the mental workload should objectively be high (landings, take-offs, cognitive tasks, etc.). The hatched region corresponds to a social task (radio communication).

**Table 1 sensors-25-03624-t001:** Features used in the model. List of all features used in the model, regrouped according to three main categories: the physiology of the operator, the state of the machine, and the interaction with the interface. Abbreviations: HR = heart rate; IBI = inter-beat interval; BR = breathing rate.

	Signal	Sampling Frequency (Hz)	Features	Unit
Physiology	ECG	256	Mean HR	bpm
			HR standard deviation	bpm
			Mean IBI	ms
			IBI standard deviation	ms
	Plethysmography	0.068±0.020	Mean BR	bpm
			BR standard deviation	bpm
	Oculometry	60	Mean fixation duration	s
			Mean saccade duration	s
			Mean saccade amplitude	deg
	Gaze (eyes + head)	60	95% prediction ellipse	pixels^2^
			Time spent in each AoI	s
Machine	Helicopter position	25	Altitude standard deviation	ft
			Altitude mean-crossings	
			Yaw orientation standard deviation	deg
			Yaw orientation mean-crossings	
			Mean pitch orientation	deg
			Pitch orientation standard deviation	deg
			Pitch orientation mean-crossings	
			Mean roll orientation	deg
			Roll orientation standard deviation	deg
			Roll orientation mean-crossings	
	Automatic pilot	Not applicable	Time spent in each horizontal sub-mode	s
			Time spent in each vertical sub-mode	s
Human–machine interface	Flight instruments: thrust lever, anti-torque pedals, cyclic pitch, cyclic roll	25	Mean displacement	%
			Displacement standard deviation	%
			Displacement mean-crossings	
			Mean force	daN
			Force standard deviation	daN
			Force zero-crossings	
	Radio communication	Not applicable	Proportion of time spent in communication	%

**Table 2 sensors-25-03624-t002:** Model performance. Median, first quartile, third quartile, mean, and standard deviation of the AUC for each pilot (1000 repetitions). The model was trained with the self-assessments as ground truth and all feature sets.

N° Pilot	ROC AUC (Median [25%, 75%])	$\mathrm{ROC}\;\mathrm{AUC}\;(\bar{m}\;\pm \;std$)
3	$0.750\;\left[0.651,\;0.833\right]$	0.690±0.227
4	$0.917\;\left[0.781,\;1.000\right]$	0.855±0.206
5	$0.875\;\left[0.780,\;1.000\right]$	0.826±0.212
6	$0.944\;\left[0.820,\;1.000\right]$	0.874±0.190
7	$0.875\;\left[0.754,\;1.000\right]$	0.837±0.187
8	$0.841\;\left[0.734,\;1.000\right]$	0.807±0.205
9	$0.833\;\left[0.714,\;1.000\right]$	0.803±0.211

**Table 3 sensors-25-03624-t003:** Best features. Features used more than 90% of the time by the model, in descending order. The model was trained with the self-assessments as ground truth and all feature sets. A complete version of this table can be found in Appendix A.

Features	Percentage of Use
Standard deviation of the displacement: pedals, cyclic in the pitch and roll planes	100, 100, 100
Standard deviation of the position of the helicopter: yaw, pitch and roll planes	100, 100, 100
Standard deviation of the force applied: pedals, cyclic in the pitch plane	100, 100
Movement frequency of the pedals	100
Proportion of time looking out of the cockpit	100
Standard deviation of the displacement and of the force applied on the collective lever	98.7, 97.6
Proportion of time spent with a horizontal or vertical autopilot	98.5, 97.5
Standard deviation of helicopter altitude	95.7
Mean heart rate	93.3

**Table 4 sensors-25-03624-t004:** Model performance on the different subsets of features. Global AUC for the different subsets of features tested. On each row, only the subset that was not shuffled is reported.

Dataset	$\mathrm{ROC}\;\mathrm{AUC}\;(\bar{m}\;\pm \;std$)	$\mathrm{Max}\;F1\;(\bar{m}\;\pm \;std$)	$\mathrm{PR}\;\mathrm{AUC}\;(\bar{m}\;\pm \;std$)
All	0.837±0.081	0.842±0.078	0.820±0.097
Physiological data	0.699±0.093	0.743±0.061	0.685±0.089
Machine data	0.777±0.100	0.784±0.083	0.762±0.108
Interface data	0.763±0.088	0.777±0.080	0.764±0.095

**Table 5 sensors-25-03624-t005:** Model performance using different ground truths. Mean ROC AUC score and standard deviation with both the experimental and theoretical NASA-TLX and their subsets as ground truth, and the whole dataset as input.

NASA-TLX Subsets	$\mathrm{ROC}\;\mathrm{AUC}\;(\bar{m}\;\pm \;std$)
RTLX (unweighted mean)	0.733±0.158
Mental demand	0.756±0.203
Temporal demand	0.617±0.167
Frustration	0.372±0.147
Effort	0.804±0.172
Theoretical RTLX (unweighted mean)	0.622±0.122
Theoretical mental demand	0.599±0.083
Theoretical temporal demand	0.512±0.108
Theoretical effort	0.615±0.110

## Data Availability

Due to the sensitive nature of military pilot data, the dataset used in this study cannot be shared publicly. However, the machine-learning model developed is available online in a public repository.

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
