# Peer review of "Continuous Assessment of Mental Workload During Complex Human–Machine Interaction: Inferring Cognitive State from Signals External to the Operatorâ€"

_sensors, 2025, doi:10.3390/s25123624_

Round 1
Reviewer 1 Report
Comments and Suggestions for Authors
In this paper, a data-driven method based on physiological and operational parameters is proposed. The mental workload ( MWL ) of professional helicopter pilots in real simulated flight is continuously estimated by machine learning model.
One of major issues is that the structure of the manuscript should be re-organized.
Another issue is that the manuscript appear too lengthy to read. It contain too much less important information that could be simplified.
Why bagging algorithm is used, as there are many other algorithms?
How the MWL is devided? How to define high and low?
The evaluation metric could be more, apart from AUC.
Some more comments:
Lines 713–714: The median value reported is 47.78, yet a threshold of 50 was adopted, citing a lack of data entries between 47 and 50. However, the rationale behind this choice remains unclear and requires further justification.
The study includes two types of aircraft tasks with mental workload controlled through task differences. It should be clarified whether variations in task type and duration influenced the task difficulty and, consequently, the workload levels.
The adequacy of using data from only nine pilots needs to be addressed. Is this sample size sufficient to ensure the reliability and generalizability of the findings?
The reported correlations between physiological features and mental workload are relatively low. The influence of the simulated environment on these results should be discussed, especially considering how such signals might vary in complex, high-altitude operational contexts.
The HBagging algorithm is chosen as the classifier, but only minimal explanation is provided regarding its noise reduction capabilities and robustness. A more detailed rationale and comparison with alternative algorithms would strengthen this section.
The study aims to detect mental workload via external signals, but ECG is used as a key input. Clarification is needed on whether ECG qualifies as an external signal in this context.
Although the study mentions the use of a simulation scenario, no actual images or visuals of the experimental setup are provided. Including such visuals would enhance the paper's transparency and reproducibility.
Please consider capitalizing the first letter in “Figure 2” to maintain consistency with standard figure caption conventions.
A number of references are over a decade old. The paper would benefit from integrating more recent studies to enhance its relevance and reflect current advancements.
Attention should be given to formatting, spelling, and overall language consistency to improve the clarity of the manuscript.
Author Response
The authors would like to express their sincere gratitude toward both reviewers. Your insightful feedback and suggestions have greatly helped us in improving the manuscript.
Please see the attached document that, hopefully, will address your concerns.

Reviewer 2 Report
Comments and Suggestions for Authors
Review on the study on “Continuous assessment of mental workload during complex 2 human-machine interaction: inferring cognitive state from sig- 3 nals external to the operator”
Dear Authors,
The paper offers an exciting technique that uses machine learning to continuously quantify mental workload in high human-machine interactions. The study is well-structured and gives useful information about the predictive potential of operational and physiological parameters for mental workload measurement. However, some significant challenges must be addressed in order to improve scientific rigor, clarify feature selection, and broaden the study's general applicability. The following are specific suggestions to improve the manuscript.
- The research's methodological approach requires better articulation and visualization. Given the various data sources and the machine learning technique applied, an organized theoretical framework would provide more clarity.
- The study's sample size of only nine helicopter pilots is insufficient for training machine-learning algorithms. This may limit the results’ general applicability. To increase robustness, consider recruiting additional participants or augmenting the research with data from more pilot training sessions.
- According to the study, operational factors predict MWL accurately than physiological signs. However, the logic for the model's feature selection could be more clearly articulated.
Author Response

(The authors gave the same response as above.)

Round 2
Reviewer 2 Report
Comments and Suggestions for Authors
Dear Authors,
Thank you for your thorough and extensive improvements. I thoroughly studied your remarks and the revised manuscript.
You thoroughly handled all reviewer suggestions, clarified methodological issues, included relevant visualisations, and enhanced the paper's clarity and structure. The manuscript's readability and scientific credibility have improved greatly as a result of the modifications.
I am pleased with the improvements and appreciate your efforts to improve the manuscript. As a result, I am happy to accept the manuscript in its current version.
Best Regards,
Reviewer
Author Response
Dear Reviewer 2,
Thank you very much for taking the time to study the new version of the manuscript.
We are pleased to hear that the modifications have improved its readability and its scientific content.
The authors would like to thank you once more for your valuable remarks and suggestions that have been of great help in improving the quality of the manuscript.
Best regards,
The authors